# A Machine Learning Approach for Investigating Delirium as a Multifactorial Syndrome

**DOI:** 10.3390/ijerph18137105

**Published:** 2021-07-02

**Authors:** Honoria Ocagli, Daniele Bottigliengo, Giulia Lorenzoni, Danila Azzolina, Aslihan S. Acar, Silvia Sorgato, Lucia Stivanello, Mario Degan, Dario Gregori

**Affiliations:** 1Unit of Biostatistics, Epidemiology and Public Health, Department of Cardiac, Thoracic, Vascular Sciences and Public Health, University of Padova, Via Loredan 18, 35121 Padova, Italy; honoria.ocagli@unipd.it (H.O.); daniele.bottigliengo@studenti.unipd.it (D.B.); giulia.lorenzoni@unipd.it (G.L.); danila.azzolina@unife.it (D.A.); 2Department of Medical Science, University of Ferrara, Via Fossato di Mortara 64B, 44121 Ferrara, Italy; 3Department of Actuarial Sciences, Hacettepe University, Ankara 06800, Turkey; aslihans@hacettepe.edu.tr; 4Health Professional Management Service (DPS) of the University Hospital of Padova, 35128 Padova, Italy; silvia.sorgato@aopd.veneto.it (S.S.); lucia.stivanello@aopd.veneto.it (L.S.); mario.degan@aopd.veneto.it (M.D.)

**Keywords:** aging, nursing, delirium, machine learning technique, random forest

## Abstract

Delirium is a psycho-organic syndrome common in hospitalized patients, especially the elderly, and is associated with poor clinical outcomes. This study aims to identify the predictors that are mostly associated with the risk of delirium episodes using a machine learning technique (MLT). A random forest (RF) algorithm was used to evaluate the association between the subject’s characteristics and the 4AT (the 4 A’s test) score screening tool for delirium. RF algorithm was implemented using information based on demographic characteristics, comorbidities, drugs and procedures. Of the 78 patients enrolled in the study, 49 (63%) were at risk for delirium, 32 (41%) had at least one episode of delirium during the hospitalization (38% in orthopedics and 31% both in internal medicine and in the geriatric ward). The model explained 75.8% of the variability of the 4AT score with a root mean squared error of 3.29. Higher age, the presence of dementia, physical restraint, diabetes and a lower degree are the variables associated with an increase of the 4AT score. Random forest is a valid method for investigating the patients’ characteristics associated with delirium onset also in small case-series. The use of this model may allow for early detection of delirium onset to plan the proper adjustment in healthcare assistance.

## 1. Introduction

Delirium is a psycho-organic syndrome characterized by an alteration in attention and consciousness, with disorganized psychic activity, fragmentation of psychic processes that appear untied and upset [1]. Delirium has a multifactorial etiology, in which internal predisposing factors (susceptibility) interact with external precipitating ones [2,3]. In contrast to dementia, delirium is often reversible with early detection and treatment of underlying causes [4]. For delirium, there are many risk factors that may change according to the characteristics of the patient. In literature, risk factors for delirium were evaluated in systematic reviews according to the type of patients. Risk of delirium is higher in very old patients [5,6]. Marquetand et al. 2020 [6] in a recent work found that very old patients require only few precipitant factors to develop delirium. Age was a risk factor also for patients after hip fracture surgery [7], vascular surgery [8], in knee and hip replacement patients [9]. Other risk factors are as follows: function dependency [7,10], hypertension [8], hearing or visual impairment [7,8], anesthetic use [11], and cognitive impairment [9,10]. Older adults following elective surgery frailty and psychotropic medication have potentially modifiable prognosis factors [12].

Delirium prevalence in the general population hospitalized is low (1–2%), but increases from 29% to 64% in elderly [13], increasing morbidity, mortality, loss of independence, length of stay, and institutionalization. Delirium prevalence also affects health care costs, which amount to over $164 billion in the United States [14]. In the Italian territory, delirium prevalence is as follows: 11% in medical wards [15], 20% in the general hospital [5], 23% in general wards [16,17] and up to 37% of patients with subsyndromal delirium [18]. Given its pervasive nature and deleterious effects, delirium surveillance is recommended for hospitalized patients [19]. Under-detection or misdiagnosis is estimated in 50–75% of delirium cases [20], and 30–40% of reported delirium episodes are preventable [21]. In addition, delirium detection is complicated by the characteristics of hospitalized patients who are often frail elderly and, therefore, more susceptible to delirium onset, especially during hospitalization where the intensification of treatments and diagnostic interventions become potentially precipitant factors [15].

DSM-V criteria are the standard for the diagnosis of delirium in clinical setting. Recently, time-efficient tools have been developed in the clinical setting for delirium detection and diagnosis [22]. In literature, there are several tools for facilitating delirium detection [22,23], such as the Confusion Assessment Method (CAM) [24], 4AT test [25], and the most recent Nursing DElirium SCreening (Nu-DESC) tool [19]. Both CAM [26,27,28] and Nu-DESC scales were used to compare the ability of random forest (RF) models in predicting the risk of delirium episodes. The structure of these tools shows similar domains. Nu-DESC has five domains: (1) disorientation, (2) inadequate behavior, (3) inadequate communication, (4) hallucination, and (5) psychomotor delay. CAM consists, instead, of four main themes: (1) acute onset and fluctuating course, (2) inattention, (3) disorganized thinking, and (4) altered level of consciousness. Bellelli [25], for the first time, used the 4AT score to assess patients at risk for delirium. Machine learning techniques (MLTs) have been widely used in data-driven prediction models, for example in dementia [29], or delirium as well with performances comparable with traditional logistic regression [26] and included into clinical workflow [30].

The study has a twofold goal: (1) to identify, using a machine learning approach, which subject’s characteristics are mostly associated with 4AT score and how they impact its variability and (2) to evaluate delirium presence/absence in older patients during the first five days of hospitalization using a standardized tool which is the 4AT instrument.

## 2. Materials and Methods

### 2.1. Study Design and Inclusion Criteria

This observational study has taken place between August and September 2016 in the Orthopedics, Geriatrics, and General Medicine wards of the University Hospital of Padova. The inclusion criteria for the enrolled patients were as follows; aged above 65 years old, understanding he Italian language and length of stay of at least five days. Patients with a psychiatric illness already diagnosed at admission, with communication problems (such as aphasia, coma status), or with a terminal disease, were excluded from the study.

An experienced nurse explained the purpose of the study and obtained informed consent from the patient or his next of kin when the participant was unable to give his consent. Study participants were evaluated with the 4AT instrument by three trained nursing students. The project was conducted within the framework of thesis preparation of the nursing student and it was approved by the internal offices at the University Hospital of Padova.

### 2.2. Variables Collected

Each patient was assessed for delirium with the 4AT scale [25] at least 24 h after the admission and each day till the fifth day of hospitalization. For each patient, the following variables were collected in different moments: (i) at hospital admission: socio-demographic characteristics such as ward, age, diagnosis, degree of study, comorbidities potentially affecting the delirium onset (dementia, alcoholism, drugs addiction, depression, other psychiatric diseases, diabetes, cancer, malnutrition); previous admission to hospital, history of delirium, hearing and visual impairment; (ii) daily variables during the hospitalization: sleep deprivation, hours of caregiver assistance, 4AT score; (iii) each shift (three times per day, morning, afternoon and night): mobility (bed transfer-chair, walking, use of stairs); physical restraint; presence/absence of invasive device (urinary catheter, peripheric venous catheter, central venous catheter, feeding tube, percutaneous endoscopic gastrostomy); pain, fever, surgery; drugs (affecting central nervous system: anticholinergic, dopaminergic, steroid, opioid, antiepileptic, anti-anxiety, neuroleptics, antidepressants); and antibiotics (quinolone, antifungal voriconazole and cephalosporins).

### 2.3. Instrument

The 4AT [25] is a short and handy instrument for routine detection of delirium and cognitive impairment by unskilled hospital personnel; on average, it takes less than two minutes to fill it. The total score ranges from 0 to 12 and is structured in 4 domains (alertness, orientation, attention and fluctuation). The compilation of items 1–3 is based exclusively on the patient’s observation at the time of the evaluation, Item 4 compilation, requires instead the collection of information from multiple sources. A score of 0 suggests that delirium and/or cognitive impairment are unlikely but do not exclude them. A score of 1–3 is suggestive of cognitive impairment and requires a more detailed cognitive examination, whereas a score higher than four suggests for delirium and requires further clinical assessment since the tool is not diagnostic. The instrument has a sensitivity of 89.7% and a specificity of 84.1% with a positive likelihood ratio of 5.62 and a negative likelihood ratio of 0.12 in the validation study [31]. However, it has shown a poor specificity, which ranges from 53.7% (95% CI: 48.1–59.2) [32] to 0.91 (0.88–0.94) [33]. The instrument was validated in the elderly population in several languages [34,35], various contexts such as the emergency department [33], hospice [36], stroke unit [37,38], medical ward [32] and in different cultural backgrounds [39]. The instrument also has some limitations; it is not suitable for patients with impaired communications and the mere presence of changes in alertness or a fluctuating course of the mental status is sufficient to define the patient at risk for delirium [35].

### 2.4. Statistical Analysis

Categorical variables were summarized according to delirium profile groups, with relative and absolute frequencies. As only categorical variables were considered, the Chi-square test was performed. The *p*-values were also reported for all possible pairwise comparisons between delirium categories. For these comparisons, the Benjamini-Hochberg adjusted *p*-values and the unadjusted *p*-values have been computed [40].

A *p*-value lower than 0.05 is conventionally considered meaningful.

The effective sample size on the observations was computed using Kish [41] formula as follows: (i) the scores of simple standard deviation (SSD) and the robust to heteroskedasticity standard deviation (RSD), were obtained using individuals as clusters; (ii) the ratio between RSD and SSD was computed; (iii) the effective sample size was retrieved as the ratio between the number of observations in the sample and the ratio computed at the previous step. Records with missing assessment of delirium were removed.

A logistic regression model was calculated to assess the time effect on the risk of developing delirium and having delirium during the hospitalization. The variance estimates were calculated by considering the Huber–White [42] estimator to account for the correlation within repeated measurements. Patients with a score higher than one were considered at risk for delirium.

### 2.5. Machine Learning Approach

An ML approach was used to evaluate the association between the subject’s characteristics and the 4AT score. MLTs can easily detect non-linear relationships and interactions and can be used with a low number of subjects, in a case where the standard statistical approaches may have some limitations [43]. The random Forest algorithm, one of the most popular methods in the MLT field [44], was implemented to describe the 4AT score based on the following set of predictors: age, gender, physical restraint, mobility, dementia, diabetes, cancer, ward, degree level, previous episodes of delirium and admission to hospital, and addiction (at least one between alcoholism, drugs addiction, depression, and other psychiatric diseases); at least one antibiotic; at least one drugs affecting the central nervous system, at least one invasive device, at least one among pain and fever, at least one among visual and hearing impairment.

The parameters of the algorithm were chosen such that the root mean squared error (RMSE), i.e., the root of the average squared difference between observed and predicted the 4AT score, was minimized. The RF algorithm was implemented using 10,000 trees.

The database is composed of repeated measurements within-subject; for this reason, the RF algorithm has been forced to use stratified sampling within each subject as indicated in the literature for cluster correlated data [45].

Variable importance using the permutation method [46] was used to assess the predictors that mostly impact the variability of the 4AT score. Partial dependence plots (PDPs) were used to depict how the 4AT score changes given the characteristics of the subjects [47]. Briefly, PDPs represent the score values predicted by the model for predictor’s value by marginalizing over the values of the other variables which were observed in the sample. PDPs are often used to aid the interpretation of an ML model by describing the relationship between a predictor and an outcome.

Analyses were performed using R software 3.6.1 (CRAN: Viena, Austria) [48]. The RF algorithm was implemented using the randomForestSRC R package (version 2.9.1) (CRAN: Viena, Austria) [49].

## 3. Results

In the study, 78 patients were enrolled, for a total of 1149 observations entered in the model and an effective sample size of 95 independent observations computed using the formula from Kish [41]. A total of 49 (63%) patients were at risk for delirium (4AT score higher than 1), 32 (41%) patients experienced delirium at least once (4AT score higher than 4) during their stay. In both cases are prevalently present in geriatrics ward, respectively 23 (96%) and 14 (58%), and in orthopedics ward, 17 (57%) and 12 (40%) respectively. Patients at risk for delirium were mainly females were 52 (64%) of the population, with a medium–low education level and an age between 80 and 90 years. A statistical description of the whole sample according to the 4AT score profile group at baseline assessment is reported in Table 1.

The logistic regression model shows a *p*-value of 0.09 indicating a non-significant effect of hospitalization time on the risk of developing delirium or cognitive impairment. The same finding has been identified (*p*-value 0.08) by considering the possible time effect on the risk of developing delirium during the hospitalization. The prevalence of patients at risk for delirium for each day varies from 53% to 60% (Figure 1, blue line). Instead, the prevalence of delirium (4AT score higher than 4) for each day varies from 19% to 29% (Figure 1, red line).

Patients with a 4AT score that suggest delirium or cognitive impairment were prevalently in geriatrics 7 (39%) and orthopedics 8 (44%) wards (*p*-value < 0.001), had mainly dementia 8 (44%, *p*-value 0.007) and were aged between 91–95 years of age 7 (39%, *p*-value 0.035) (Table 1).

The RF model explained 75.8% of the variability of the 4AT score with an RMSE of 3.29, i.e., the average difference between the observed and predicted 4AT values is 3.29 points. Figure 2 shows the ranking of the predictors according to the importance attributed by the algorithm measured by the associated relative decrease in the model’s predictive error.

For example, when age is considered, the algorithm’s predictive error decreases by 43.1%. Age, the presence of physical restraint, dementia, type of ward, educational level, and gender, were the variables most associated with the 4AT score. Table 2 shows the predicted median (I and III quartiles) 4AT score values for each relevant variable marginalized over all the other variables. The 4AT predicted score values were obtained using the PDPs approach.

For example, if suffering from dementia, the RF algorithm predicts a median 3.72 4AT score (3.69 as I Quartile and 3.79 as III Quartile), whereas a median 2.36 4AT score (2.32 as I Quartile and 2.42 as III Quartile) is predicted by the model if the subject did not suffer from dementia. The subject’s characteristics that increase the 4AT score are higher age (2.59 [2.5–2.68]), the presence of dementia (3.72 [3.69–3.79]), physical restraint (3.2 [3.15–3.28]), diabetes (3.14 [3.07–3.2]), and a lower degree. Moreover, patients in the geriatrics and medicine ward, along with patients that have more than one antibiotic and previous admission to the hospital, are at higher risk of facing delirium onset.

Figure 3 and Figure 4 report the effect of the variables on the 4AT score obtained using the PDPs approach. For example, a patient aged 92 years of age report an estimated 4AT score of 7 (Figure 3).

## 4. Discussion

Our findings are consistent with those existing in the literature [24,50], except those related to prevalence. It should be noted that other studies have mainly focused on incidence rather than prevalence [24,51]. Our patients show a higher prevalence of delirium (41%) compared to those of Bellelli [5] (22.9%) and Zuliani [18] (37%), which are studies conducted in similar settings. This difference in the prevalence could be explained by the assessment of delirium in our study in wards such as orthopedics, geriatrics, and general medicine where risk factors for delirium are commonly present [52]. Our results, however, when considering patients at risk for delirium (63%), are similar to those reported in the study of Gehrke in patients older than 90 years in an internal medicine ward (69%) [53] and in a Colombian study in a geriatric unit where the prevalence was 51.03% [54].

As in the study of Belelli, [5], in our RF model, delirium is associated with older age, gender, use of antibiotics, physical restraints, and ward of admission. In this study, the variables that influence the increase of the 4AT score are the ones chosen from the models and already considered as predictors of delirium in other studies [3,5,55]. However, some of the factors such as predisposing ones (e.g., disability) and the precipitating factors (use of drugs active on SNC, peripheral venous and urinary catheters) do not increase the 4AT score in our group.

Several studies have tried to identify risk factors both with traditional statistical methods [55,56] and with MLT [57,58,59,60]. The studies on MLT usually use patients’ characteristics retrieved in healthcare records only in the first 24/48 h after admission. In our study, we have developed the RF algorithm accounting for all the patient’s records observed from the admission to the end of the study. In literature, age is one of the most predictive variables in delirium onset using MLT, regardless of the kind of algorithm [28,57,59,61] as also shown in our results.

Recent applications of the ML technique in predicting the risk factors of delirium onset have shown good results in favor of these techniques. In the study of Wong [59], delirium onset was evaluated using different MLTs, i.e., RF, artificial neural networks, etc., and showed that all ML methods outperform the Nu-Desc clinical tool for delirium risk assessment. The authors observed that gradient boosting machine (GBM) shows the best predictive performances with an AUC of 0.855, suggesting a good predictive ability. Davoudi [62] also, compared different ML approaches in delirium detection risk factors, showing that RF and generalized additive models (GAM) were the ones that best predict it with an AUC of 0.85 (0.83–0.86) and 0.86 (CI: 0.84–0.88) respectively. In the study of Corradi [27] instead, the RF model was the model used for predicting delirium episodes with an AUC of 0.909 (95% CI 0.898 to 0.921). All the models presented in these studies show better performance when compared to a clinical tool. Another recent study have developed a predictive model for post-operative delirium in older surgical patients with better performances than chance, but with similar performances when compared with the traditional stepwise logistic regression [26]. Another recent study has showed the high predictive ability of RF model in detection delirium based on CAM and delirium observation screening scale (DOSS) [28].

### Limitations

The present results cannot be generalizable to the whole hospital since the study has taken place in wards, where risk factors for delirium onset are higher compared to the other settings. Furthermore, data are limited thus avoiding the possibility of establishing a setting-specific prevalence of delirium. In this study, the 4AT score was used over several days, although it is suggested to submit it only once for delirium detection. The choice to adopt this tool derives from its simplicity of use and it allows comparison with other studies on the prevalence of delirium in different health contexts. From a statistical point of view, the low number of subjects enrolled in the study and the absence of an external dataset to validate the algorithm performances may limit the generalizability of the findings and the reliability of the model.

The present results show that MLT can identify complex relationship among data, so in the future it would be interesting to use those techniques to define delirium risk factors instead of traditional approaches. Moreover, it would be important to consider delirium as a “dynamic” condition influenced by different factors that are working in different moments as also theorized by Fan et al. [60] and applied in a recent work of our group [61].

## 5. Conclusions

Early detection of delirium must be a purpose to pursue in wards with a high percentage of risk factors, to adequately treat them and to avoid side effects. The routine uses of a simple instrument, such as the 4AT scale, could help to increase the awareness of delirium among health care workers. Healthcare personnel, especially nurses, thanks to their strictly engagements with patients, must be trained to easily recognize risk factors for delirium. As hypothesized in the recent work of Oberai et al. 2021 [57], nurses require educational intervention that include a variety of teaching style. Moreover, our results, following existing literature, shows that to predict correctly delirium is more useful to use predictive models that consider many factors collected routinely. These methods may also help in small case-series even if they are thought to be useful for large data sets. MLT can identify complex relations between variables and are helpful in structuring predictive models to personalize healthcare assistance.

## Figures and Tables

**Figure 1 ijerph-18-07105-f001:**
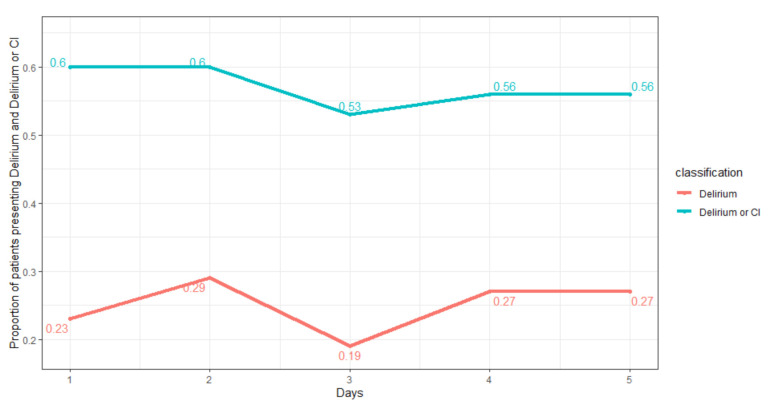
Proportions of patients reporting delirium and risk for delirium during the first five days of hospitalization. The delirium state indicates a 4AT score higher than 4; the delirium or cognitive impairment (CI) state indicates a 4AT score higher than 1.

**Figure 2 ijerph-18-07105-f002:**
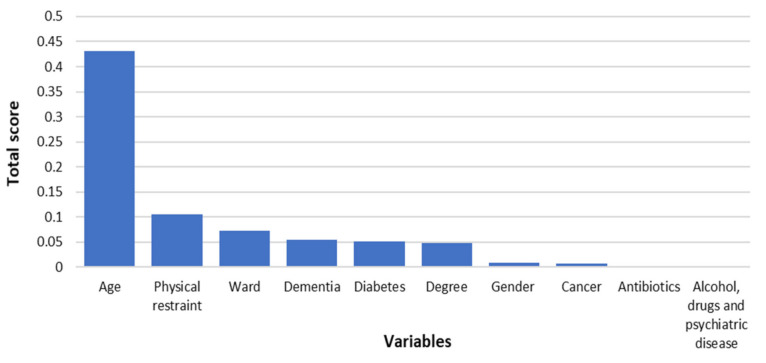
Variables with top importance selected by the RF algorithm according to the permutation approach. The total score measures the predictive impact of the variables, i.e., the relative decrease of the algorithm’s predictive error produced by a variable.

**Figure 3 ijerph-18-07105-f003:**
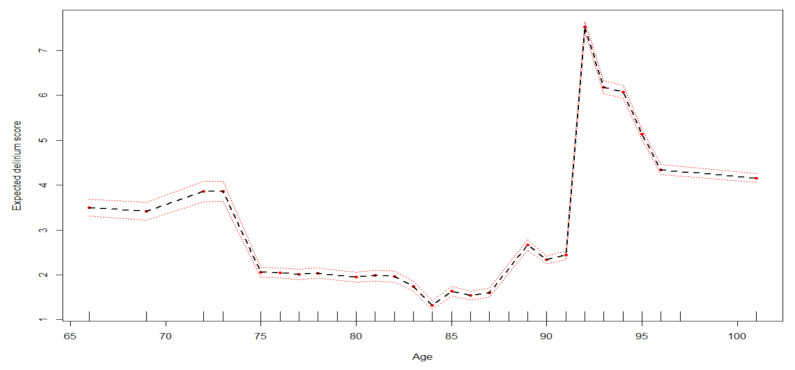
Effect of age on the 4AT delirium score. Expected delirium score estimated with random forest has been reported on the y axis according to ages with 95% confidence bounds.

**Figure 4 ijerph-18-07105-f004:**
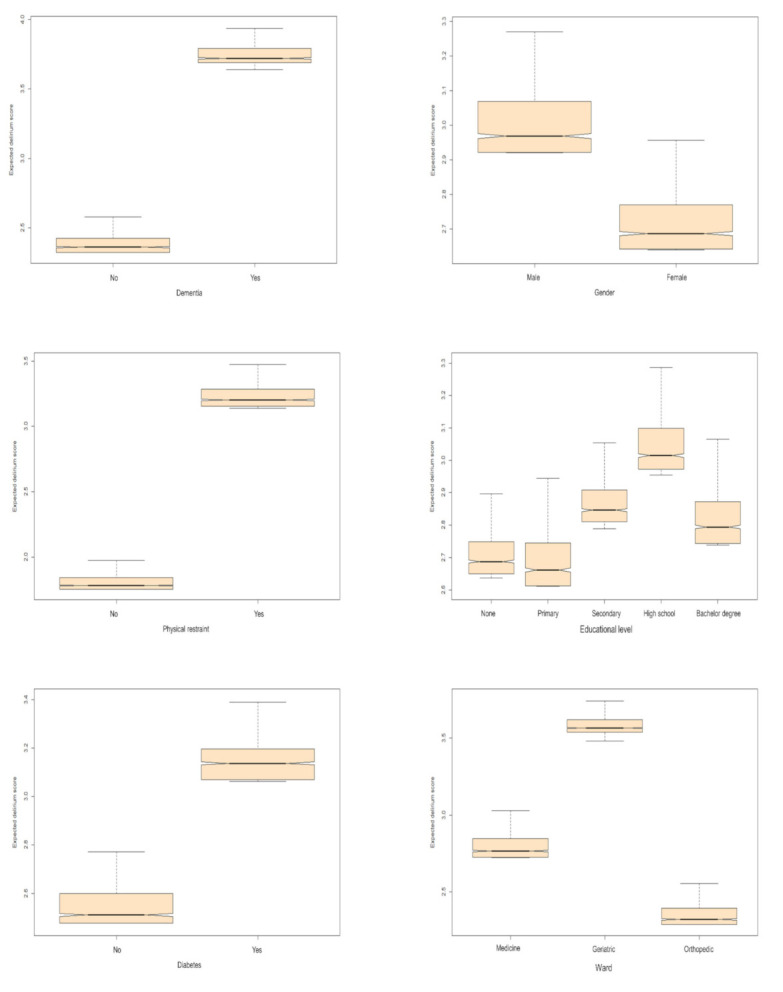
Effect of the presence of dementia, gender, physical restraint, educational level, diabetes, ward, antibiotics, and previous admissions on the 4AT delirium score. The vertical axis displays the ensemble expected predicted delirium score.

**Table 1 ijerph-18-07105-t001:** Descriptive statistics of the whole sample according to the 4AT score profile group at baseline assessment. Categorical variables were summarized as relative and absolute frequencies. Pearson Chi-square test was used to assess differences across 4AT scale categories.

Variable	Variable Level	*n*	Delirium or Severe Cognitive Impairment Unlikely	Possible Cognitive Impairment	Possible Delirium +/− Cognitive Impairment	Overall	*p*-Value	Unadjusted Pairwise *p*-Value			Adjusted Pairwise *p*-Value		
								PCI vs. D/CI	PCI vs. PD	D/CI vs. PD	PCI vs. D/CI	PCI vs. PD	D/CI vs. PD
			(*n* = 31)	(*n* = 28)	(*n* = 18)	(*n* = 77) *							
Ward	Medicine	78	16 (52%)	5 (18%)	3 (17%)	24 (31%)	<0.001	0.235	<0.001	0.004	0.235	0.003	0.006
	Geriatric		1 (3%)	16 (57%)	7 (39%)	24 (31%)							
	Orthopedic		14 (45%)	7 (25%)	8 (44%)	29 (38%)							
Gender	Male	78	12 (39%)	8 (29%)	6 (33%)	26 (34%)	0.71	0.746	0.161	0.305	0.746	0.4575	0.4575
ICD diagnosis	Circulatory system	35	0 (0%)	1 (10%)	0 (0%)	1 (3%)	0.17	0.259	0.135	0.32	0.32	0.32	0.32
	Musculoskeletal system		14 (93%)	7 (70%)	8 (89%)	29 (85%)							
	Digestive system		0 (0%)	0 (0%)	1 (11%)	1 (3%)							
	Respiratory		1 (7%)	0 (0%)	0 (0%)	1 (3%)							
	Undefined		0 (0%)	2 (20%)	0 (0%)	2 (6%)							
Educational level	Bachelor’s degree	78	2 (6%)	0 (0%)	1 (6%)	3 (4%)	0.24	0.533	0.044	0.262	0.533	0.132	0.393
	None		0 (0%)	5 (18%)	2 (11%)	7 (9%)							
	Missing		1 (3%)	0 (0%)	0 (0%)	48 (63%)							
	Primary school		23 (74%)	14 (50%)	11(61%)	13 (17%)							
	Secondary school		4 (13%)	7 (24%)	2 (11%)	5 (7%)							
	High school		1 (3%)	2 (7%)	2 (11%)	18 (23%)							
Dementia		78	2 (6%)	8 (29%)	8 (44%)	1 (1%)	0.007	0.181	0.028	0.001	0.181	0.042	0.003
Alcohol use		78	1 (3%)	0 (0%)	0 (0%)	5 (6%)	0.47		0.329	0.454		0.454	0.454
Depression		78	2 (6%)	3 (11%)	0 (0%)	16 (21%)	0.35	0.17	0.586	0.285	0.4275	0.586	0.4275
Diabetes		78	7 (23%)	4 (14%)	5 (28%)	29 (38%)	0.52	0.197	0.379	0.601	0.5685	0.5685	0.601
Cancer		78	10 (32%)	11 (39%)	8 (44%)	71 (92%)	0.68	0.544	0.645	0.311	0.645	0.645	0.645
Previous hospital admission		78	27 (87%)	27 (96%)	17 (94%)	29 (38%)	0.38	0.696	0.185	0.446	0.696	0.555	0.669
Visual impairment		78	13 (42%)	7 (25%)	9 (50%)	29 (38%)	0.19	0.048	0.144	0.464	0.144	0.216	0.464
Hearing impairment		78	8 (26%)	10 (36%)	11 (61%)	14 (18%)	0.047	0.17	0.313	0.024	0.255	0.313	0.072
Antibiotics	>1	78	6 (19%)	8 (28%)	0 (0%)	5 (6%)	0.071	0.02	0.45	0.05	0.05	0.45	0.08
age (classes)	<70	78	4 (13%)	1 (4%)	0 (0%)	3 (4%)	0.035	0.108	0.127	0.039	0.127	0.127	0.117
	>95		1 (3%)	0 (0%)	2 (11%)	5 (6%)							
	71–75		3 (10%)	0 (0%)	2 (11%)	11 (14%)							
	76–80		6 (19%)	4 (14%)	1 (6%)	19 (25%)							
	81–85		8 (26%)	8 (29%)	3 (17%)	21 (27%)							
	86–90		8 (26%)	10 (36%)	3 (17%)	13 (17%)							
	91–95		1 (3%)	5 (18%)	7 (39%)								

*n*: Reports the number of patients in which calculations were made. * One patient had no assessment in the 4AT score at baseline. Abbreviations: PCI: possible cognitive impairment, D/CI: delirium or severe cognitive impairment unlikely, possible cognitive impairment, PD: possible delirium +/− cognitive impairment.

**Table 2 ijerph-18-07105-t002:** Description of variables’ effect on the 4AT delirium score. The “4AT score” column reports the median (I and III quartiles) 4AT delirium score predicted by the model conditional on the variable’s values.

Variable		4AT Score
Age	78	2.09 [1.97–2.2]
	84	1.35 [1.25–1.44]
	87	1.58 [1.49–1.68]
	91	2.59 [2.5–2.68]
Dementia	No	2.36 [2.32–2.42]
	Yes	3.72 [3.69–3.79]
Gender	Female	2.69 [2.64–2.77]
	Male	2.97 [2.92–3.07]
Physical restraint	No	1.78 [1.75–1.84]
	Yes	3.2 [3.15–3.28]
Educational level	Bachelor’s degree	2.79 [2.74–2.87]
	None	2.69 [2.65–2.75]
	Primary school	2.66 [2.61–2.74]
	Secondary school	2.85 [2.81–2.91]
	High school	3.01 [2.97–3.1]
Diabetes	No	2.51 [2.48–2.6]
	Yes	3.14 [3.07–3.2]
Ward	Medicine	2.77 [2.73–2.85]
	Geriatric	3.57 [3.54–3.62]
	Orthopedic	2.32 [2.29–2.4]
Cancer	No	2.78 [2.73–2.86]
	Yes	2.78 [2.73–2.86]
Antibiotics	<1	2.53 [2.49–2.61]
	≥1	2.87 [2.82–2.95]
Previous hospital admission	No	2.73 [2.69–2.82]
	Yes	2.71 [2.67–2.8]
Alcohol, drugs and psychiatric disease	<1	2.72 [2.68–2.82]
	≥1	2.8 [2.76–2.89]

## Data Availability

The data presented in this study are available on request from the corresponding author. The data are not publicly available due to privacy.

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
