# Peer review of "A Machine Learning Approach for Investigating Delirium as a Multifactorial Syndrome"

_ijerph, 2021, doi:10.3390/ijerph18137105_

Round 1
Reviewer 1 Report
Thank you for the opportunity to review this manuscript.
Abstract
-Please remove the abbreviation from the abstract and revise it so that readers can understand it well.
Introduction
-It is necessary to add a review of the recent literature on risk factors of delirium.
Methods
-Who assessed the study participants using the 4AT instrument?
-If there were several raters, were there any training sessions on the use of the 4AT instrument for them?
Discussion
-What is the researcher's suggestion for further research?
Conclusion
- Based on the results of this study, what implications can the researcher suggest in education, research and practice regarding delirium assessment?
Author Response
Reviewer 1
Open Review
(x) I would not like to sign my review report
( ) I would like to sign my review report
English language and style
( ) Extensive editing of English language and style required
( ) Moderate English changes required
(x) English language and style are fine/minor spell check required
( ) I don't feel qualified to judge about the English language and style
|
Yes |
Can be improved |
Must be improved |
Not applicable |
|
|
Does the introduction provide sufficient background and include all relevant references? |
( ) |
(x) |
( ) |
( ) |
|
Is the research design appropriate? |
(x) |
( ) |
( ) |
( ) |
|
Are the methods adequately described? |
(x) |
( ) |
( ) |
( ) |
|
Are the results clearly presented? |
(x) |
( ) |
( ) |
( ) |
|
Are the conclusions supported by the results? |
( ) |
(x) |
( ) |
( ) |
Comments and Suggestions for Authors
Thank you for the opportunity to review this manuscript.
Thank you for the positive consideration reserved to our work.
Abstract
-Please remove the abbreviation from the abstract and revise it so that readers can understand it well.
Thanks for the suggestion, removed the unnecessary abbreviations.
Introduction
-It is necessary to add a review of the recent literature on risk factors of delirium.
Thanks for the suggestion. Added few lines on risk factors for delirium (pp 1-2).
Methods
-Who assessed the study participants using the 4AT instrument?
The participants were assessed by three trained nursing students. Added some lines (pag 2)
-If there were several raters, were there any training sessions on the use of the 4AT instrument for them?
The students were trained by two senior nurses’ who are expert in the field of delirium and part of the hospital group for delirium prevention and treatment.
Discussion
-What is the researcher's suggestion for further research?
Thanks for pointing this out. Further research on the theme may focus on the use of machine learning technique extensively thanks its ability to identify complex relations between variables. Moreover delirium and considering delirium as a non-static condition. Added few lines at the end of the discussion (page 10)
Conclusion
- Based on the results of this study, what implications can the researcher suggest in education, research and practice regarding delirium assessment?
In clinical practice we enhance the use of screening tools to evaluate delirium as also reported in the conclusion (pag 10). For what concern education we proposed that healthcare workers, especially nurses, have to be trained to early recognize risk factors for delirium since their role in daily routine assistance. In research, as added in discussion and conclusion, we suggest to implement the use of machine learning techniques since their ability to identify complex relationship among data.. Added few lines in the manuscript (pp 10-12)
Submission Date
26 April 2021
Date of this review
13 May 2021 03:58:15
Reviewer 2 Report
Prediction of delirium in elderly patients is very important as it prepares medical staff to cope with delirium in advance. I think this study is meaningful as a study that presented the usefulness of predicting delirium of the machine learning approach.
The study has a twofold goal: (1) to identify, using a machine learning approach, 66 which subject’s characteristics are mostly associated with 4AT score and how they impact its variability and (2) to evaluate delirium presence/absence in older patients during the first five days of hospitalization using a standardized tool which is the 4AT instrument.
Research data includes mobility, pain, fever, etc., which may have changed within 5 days of hospitalization, in addition to data collected by nurses on a 4AT scale. It is necessary to clarify how the 1149 obsevations presented in the results were collected over the 5 days and reflected in the prediction results by ML.
The results of this study are predicted in the ML approach of the 4AT score according to the characteristics of the subjects. However, since the presented results (Table 2, Figure 2, Figure 3) are predicted scores, I think that data that can compare the predicted values with the actual results should be presented. Presenting the actual figures and forecasts as two variables in one picture will help readers understand.
Researchers describe that the 4AT scale is not a diagnostic scale, so it only represents the level of risk. However, delirium is very important whether or not the patient has delirium. Providing actual results for whether delirium has occurred will serve the second purpose of this study. It is true that if the risk of delirium is high, the actual incidence is high, but the age and presence of dementia presented in this study are known variables that are highly related to delirium.
The abstract suggests that 49 out of 77 subjects (63%) experienced delirium at least once during hospitalization. Does the subject's experience of delirium mean for the total length of hospital stay? I don't think there are enough results and discussions for the second purpose of this study. Therefore, the author needs to make up for this.
And please clarify what the following sentence means. “The project was conducted within the framework thesis preparation of the nursing student”
Please refer to the corrections below.
- Table 1. The difference between the three groups was well measured with the Kruskal-Wallis test, but only to the extent that there is a difference between groups. It is necessary to present the results of the post hoc method of Bonferroni et al. to confirm the significance of any differences between which groups.
- This study is '77 patients were enrolled', and the N in Table1. is 78. Please confirm.
- Unification is required in the format of N number and% in parentheses. Please present the total of N and the number of each variable.
Example: 4page: 49 (63%) were at risk for delirium...
Table 1 also changed to 16 (52) instead of 52% (16)
- The variables in Table1, Table2, and Figure 3 are written in different terms. It must be unified.
Ex) Table 1 and Figure 3 are marked with degrees, and Table 2 is the Bachelor degree.
Table 1 and Figure 3 show'Ward-Medicine', so that Table 2 is internal medicine.
- Change the text prevalently in geriatrics (7,39%) to 7 (39%) on page5, and correct other texts as well.
- It is necessary to add the contents of antibiotics among the variables in Table 1.
- page 7 Figure3. Spelling correction, Physical restrain-> restraint
- Figure 3. On the x-axis of Antibiotics, ‘>1’ should be changed to ‘<1’.
Author Response
Reviewer 2
Open Review
(x) I would not like to sign my review report
( ) I would like to sign my review report
English language and style
( ) Extensive editing of English language and style required
( ) Moderate English changes required
( ) English language and style are fine/minor spell check required
(x) I don't feel qualified to judge about the English language and style
|
Yes |
Can be improved |
Must be improved |
Not applicable |
|
|
Does the introduction provide sufficient background and include all relevant references? |
(x) |
( ) |
( ) |
( ) |
|
Is the research design appropriate? |
( ) |
(x) |
( ) |
( ) |
|
Are the methods adequately described? |
( ) |
(x) |
( ) |
( ) |
|
Are the results clearly presented? |
( ) |
( ) |
(x) |
( ) |
|
Are the conclusions supported by the results? |
( ) |
( ) |
(x) |
( ) |
We thank the reviewer for the careful consideration and overall positive judgement given to our work.
Prediction of delirium in elderly patients is very important as it prepares medical staff to cope with delirium in advance. I think this study is meaningful as a study that presented the usefulness of predicting delirium of the machine learning approach.
The study has a twofold goal: (1) to identify, using a machine learning approach, 66 which subject’s characteristics are mostly associated with 4AT score and how they impact its variability and (2) to evaluate delirium presence/absence in older patients during the first five days of hospitalization using a standardized tool which is the 4AT instrument.
Research data includes mobility, pain, fever, etc., which may have changed within 5 days of hospitalization, in addition to data collected by nurses on a 4AT scale. It is necessary to clarify how the 1149 obsevations presented in the results were collected over the 5 days and reflected in the prediction results by ML.
Each patient was assessed for delirium 3 times per day for five days. So, after the removal of observations without delirium assessment, we consider for the analysis 1149 observations. Clarified in the manuscript (page 3).
The results of this study are predicted in the ML approach of the 4AT score according to the characteristics of the subjects. However, since the presented results (Table 2, Figure 2, Figure 3) are predicted scores, I think that data that can compare the predicted values with the actual results should be presented. Presenting the actual figures and forecasts as two variables in one picture will help readers understand.
We thank the reviewer for the comment. The results presented in Table 2, Figure 2, and Figure 3 described the 4AT delirium score values predicted by the ML for a given predictor while keeping constant all the other variables to the values observed in the sample. The purpose of these results is to aid the interpretation of the ML model by describing the relationship between a predictor and the 4AT score delirium learned by the model and not to compare the predicted and observed score values. Predicted and observed score values have been compared using root mean squared error (RMSE), that is the average difference between the actual and predicted scores. We added more details on the results in Table 2, Figure 2, and Figure 3 in the 2.5 section to further clarify this point.
Researchers describe that the 4AT scale is not a diagnostic scale, so it only represents the level of risk. However, delirium is very important whether or not the patient has delirium. Providing actual results for whether delirium has occurred will serve the second purpose of this study. It is true that if the risk of delirium is high, the actual incidence is high, but the age and presence of dementia presented in this study are known variables that are highly related to delirium.
The 4AT scale is an instrument to detect delirium, however, conversely to other instrument, it was structured to be used by unskilled personnel. In this sense it’s a screening tool as also considered in the systematic review of Van Velthuijsen et al 2016. The reviewer is right when suggest that older age and presence of dementia are known risk factors for delirium.
The abstract suggests that 49 out of 77 subjects (63%) experienced delirium at least once during hospitalization. Does the subject's experience of delirium mean for the total length of hospital stay? I don't think there are enough results and discussions for the second purpose of this study. Therefore, the author needs to make up for this.
Thanks for pointing this out. The statistics reported are related of delirium assessed in the first day of admission.We have implemented the second purpose of our study including the prevalence for each day. Added few lines on pages 3-4.
Figure 1 Proportion of patients presenting Delirium During the first five days of hospitalization.
A Logistic Regression model was calculated to assess the time effect on the risk of developing delirium during hospitalization. The variance estimates were calculated by considering the Huber–White estimator to account for the correlation within repeated measurements. A P-value of 0.09 indicates a non-significant effect of time of hospitalization on the risk of developing delirium.
And please clarify what the following sentence means. “The project was conducted within the framework thesis preparation of the nursing student”
The 4AT score, at the time of the study, was not used routinely in the wards of the study. The introduction of this instrument was thanks to a group of research within the hospital whose purposed is the creation of guidelines for delirium prevention. So, three nurses student for their thesis work on the introduction of this instrument as an instrument to prevent delirium episodes.
Please refer to the corrections below.
- Table 1. The difference between the three groups was well measured with the Kruskal-Wallis test, but only to the extent that there is a difference between groups. It is necessary to present the results of the post hoc method of Bonferroni et al. to confirm the significance of any differences between which groups.
As suggested by the reviewer p-value is also reported for all possible pairwise comparisons between delirium categories. For these comparisons, the Benjamini-Hochberg adjusted p-values and the unadjusted p-values are reported. The Chi-Square test was performed as only categorical variables considered in the descriptive table.
- This study is '77 patients were enrolled', and the N in Table1. is 78. Please confirm.
Checked, modified,
- Unification is required in the format of N number and% in parentheses. Please present the total of N and the number of each variable.
Example: 4page: 49 (63%) were at risk for delirium...
Table 1 also changed to 16 (52) instead of 52% (16)
Thanks for the observation, unified the format as required.
- The variables in Table1, Table2, and Figure 3 are written in different terms. It must be unified.
Ex) Table 1 and Figure 3 are marked with degrees, and Table 2 is the Bachelor degree.
Table 1 and Figure 3 show'Ward-Medicine', so that Table 2 is internal medicine.
Thanks for the observation, unified the terms in the tables and figure.
- Change the text prevalently in geriatrics (7,39%) to 7 (39%) on page5, and correct other texts as well.
Modified
- It is necessary to add the contents of antibiotics among the variables in Table 1 Thanks for the suggestion, added in table 1.
- page 7 Figure3. Spelling correction, Physical restrain-> restraint
Thanks, modified.
- Figure 3. On the x-axis of Antibiotics, ‘>1’ should be changed to ‘<1’.
Thanks, modified.
Submission Date
26 April 2021
Date of this review
27 May 2021 04:51:38
Round 2
Reviewer 2 Report
The abstract states:
“Of the 787 patients enrolled in the study, 49 (63%) had at least one episode of delirium during the hospitalization (38% in orthopedics and 31% both in internal medicine and in the geriatric ward).”
However, the text states:
“ 49 (63%) patients were at risk for delirium (38% in orthopedics and 31% both in 183 internal medicine and in the geriatric ward) in the first day of admission”
In this article, 49 patients are presented as subjects with a high risk of delirium at the time of admission. Are they subjects who experienced delirium during hospitalization as suggested in the abstract? Also, if the number of study participants was 787, how did 49 (63%) come out?Please make this clear.
Whether the participants in this study were 77.Please clarify whether it is 78 or 95 in the table or description. The text states as follows:
“In the study, 787 patients were enrolled, for a total of 1149 observations entered in the model and an effective sample size of 95 subjects computed using the formula from 182 Kish [42]. 49 (63%) patients were at risk for delirium (38% in orthopedics and 31% both in 183 internal medicine and in the geriatric ward) in the first day of admission, for a total of 1149 observations entered in the model and an effective sample size of 95 subjects computed using the formula from Kish [42]. Females were 52 (64%) of the population, with a medium-low education level and an age between 80 and 90 years. A statistical description of the whole sample according to the 4AT score profile group is reported in Table 1. “
And in Table 1, describe the total number of N, and classify the number of participants in Medicine, Geriatric, and Orthopedic respectively.Table 1 shows that there were 78 male participants.It is also necessary to clarify the number of participants for ICD diagnosis and other subject characteristics.
Author Response
Open Review
(x) I would not like to sign my review report
( ) I would like to sign my review report
English language and style
( ) Extensive editing of English language and style required
( ) Moderate English changes required
( ) English language and style are fine/minor spell check required
(x) I don't feel qualified to judge about the English language and style
|
Yes |
Can be improved |
Must be improved |
Not applicable |
|
|
Does the introduction provide sufficient background and include all relevant references? |
( ) |
(x) |
( ) |
( ) |
|
Is the research design appropriate? |
( ) |
(x) |
( ) |
( ) |
|
Are the methods adequately described? |
( ) |
(x) |
( ) |
( ) |
|
Are the results clearly presented? |
( ) |
( ) |
(x) |
( ) |
|
Are the conclusions supported by the results? |
( ) |
(x) |
( ) |
( ) |
Comments and Suggestions for Authors
The abstract states:
“Of the 787 patients enrolled in the study, 49 (63%) had at least one episode of delirium during the hospitalization (38% in orthopedics and 31% both in internal medicine and in the geriatric ward).”
However, the text states:
“ 49 (63%) patients were at risk for delirium (38% in orthopedics and 31% both in 183 internal medicine and in the geriatric ward) in the first day of admission”
In this article, 49 patients are presented as subjects with a high risk of delirium at the time of admission. Are they subjects who experienced delirium during hospitalization as suggested in the abstract?
Thanks for pointing this out. 49 are the subjects that are at risk for delirium at least once during their stay according to the 4AT scale, 32 (41,5%) had at least one episode of delirium in the same period. Patients at risk for delirium are those with a score higher than one, instead patients that have experienced delirium had a score higher than four. Clarified in the manuscript.
Table 1 report the statistics related to the baseline observations for each patient.
Also, if the number of study participants was 787, how did 49 (63%) come out?Please make this clear
The study participants were 78, in the baseline assessment there was a missing data for the 4AT scale, so in table 1 were reported 77 patients in the column related to the score. In this study we have considered patients at risk for delirium when the score was higher than 1, so 49 (63%). Patients that experienced at least one episode of delirium were those with the 4AT score higher than 4. Clarified in the manuscript.
Whether the participants in this study were 77.Please clarify whether it is 78 or 95 in the table or description. The text states as follows:
Patients enrolled were 78, in table 1 the data related to the score are reported for 77 patients since for baseline assessment there was a missing data for the 4AT score. Clarified in the manuscript. 95 is the number of independent observations computed with Kish formula which reflects the effective sample size after accounting for the repeated measurements for each subject.
“In the study, 787 patients were enrolled, for a total of 1149 observations entered in the model and an effective sample size of 95 subjects computed using the formula from 182 Kish [42]. 49 (63%) patients were at risk for delirium (38% in orthopedics and 31% both in 183 internal medicine and in the geriatric ward) in the first day of admission, for a total of 1149 observations entered in the model and an effective sample size of 95 subjects computed using the formula from Kish [42]. Females were 52 (64%) of the population, with a medium-low education level and an age between 80 and 90 years. A statistical description of the whole sample according to the 4AT score profile group is reported in Table 1. “
And in Table 1, describe the total number of N, and classify the number of participants in Medicine, Geriatric, and Orthopedic respectively.Table 1 shows that there were 78 male participants.It is also necessary to clarify the number of participants for ICD diagnosis and other subject characteristics.
Thanks for the suggestion. Added the column overall in table 1 to show the number of participants for each level of the categorical variables. Maintained the column N that shows the number in which the statistics were computed.
Submission Date
26 April 2021
Date of this review
07 Jun 2021 08:51:26